materials science/nanotechnology/green chemistry

black phosphorus, phosphorene, liquid-phase exfoliation, atomic force microscopy, Raman spectroscopy

**Authors for correspondence:**
Amine El Moutaouakil
e-mail: a.elmoutaouakil@uaeu.ac.ae
Omar Mounkachi
e-mail: omar.mounkachi@um5.ac.ma

This article has been edited by the Royal Society of Chemistry, including the commissioning, peer review process and editorial aspects up to the point of acceptance.

# Efficient production of few-layer black phosphorus by liquid-phase exfoliation

Ghassane Tiouitchi[1], Mustapha Ait Ali[2], Abdelilah Benyoussef[1,3], Mohammed Hamedoun[1], Abdessadek Lachgar[4], Abdelkader Kara[5], Ahmed Ennaoui[6], Abdelfattah Mahmoud[7], Frederic Boschini[7], Hamid Oughaddou[8,9], Amine El Moutaouakil[10], Abdellah El Kenz[1] and Omar Mounkachi[1]

[1]Laboratory of Condensed Matter and Interdisciplinary Sciences (LaMCScI), BP 1014, Faculty of Science-Mohammed V University, Rabat, Morocco
[2]Coordination Chemistry Laboratory, Cadi Ayyad University, Faculty of Sciences Semlalia (UCA-FSSM), BP 2390-40000 Marrakech, Morocco
[3]Hassan II Academy of Science and Technology, Rabat, Morocco
[4]Department of Chemistry, Wake Forest University, Winston-Salem, NC, USA
[5]Department of Physics, University of Central Florida, Orlando, FL 32816, USA
[6]Institut de Recherche en Energie Solaire et Energies Nouvelles, Rabat, Morocco
[7]GREENMAT, CESAM, Institute of Chemistry B6, University of Liege, 4000 Liège, Belgium
[8]Institut des Sciences Moléculaires d'Orsay, ISMO-CNRS, Bât. 210, Université Paris-Sud, 91405 Orsay, France
[9]Département de Physique, Université de Cergy-Pontoise, 95031 Cergy-Pontoise Cedex, France
[10]Department of Electrical Engineering, College of Engineering, UAE University, Al Ain, United Arab Emirates

AEM, 0000-0002-7407-020X

Phosphorene is a new two-dimensional material that has recently attracted much attention owing to its fascinating electrical, optical, thermal and chemical properties. Here, we report on high-quality exfoliation of black phosphorus nanosheets, with controllable size produced in large quantities by liquid-phase exfoliation using *N*-methyl-2-pyrrolidone (NMP) as a solvent under ambient conditions. The as-synthesized few layers show a great potential for solar energy conversion based on the optical results shown in this work.

# 1. Introduction

In recent years, the study and investigation of two-dimensional (2D) materials became among the most attractive and exciting aspects of nanoscience. Phosphorene is a 2D allotrope of phosphorus. Similar to graphene layers that stack together to form graphite, phosphorene monolayers can be stacked via van der Waals interactions to build crystal layers of black phosphorus. However, black phosphorus is a semiconductor with a direct band gap in single, few-layer and bulk forms. The direct band gap depends on the nanosheet thickness; its value goes from approximately 1.5 eV for a monolayer phosphorene, reaching approximately 0.3 eV for bulk black phosphorus [1,2], unlike graphene that has no band gap [3] and $MoS_2$ that displays direct band gaps only in the monolayer forms [4]. Thanks to its intrinsic band gap, black phosphorus is considered a suitable semiconductor for use in gas sensors [5,6], photovoltaic applications [7], solar cells, energy storage [8], and electronic [9–11] and optical devices [12–16]. The exfoliation energy of black phosphorus is −151 meV per atom, calculated by multi-level quantum chemical calculations [17], which is greater than that of graphite. This presents a relatively major difficulty in exfoliating black phosphorus. The scotch-tape method or mechanical exfoliation, which is also known for being used in the case of graphene and transition metal dichalcogenides (TMDCs), produces high-quality single and multi-layer phosphorene sheets. However, only small-sized crystals are obtained using this process that is still limited to laboratory scale. On the other hand, liquid-phase exfoliation (LPE) [18], based on the ultrasonic exfoliation of black phosphorus, produces colloidal dispersions of nanosheets in a solution, and can be considered as a better process for mass production. The centrifugation with different speeds can separate particles with different sizes in the phosphorene from LPE. Through this method, black phosphorus is exfoliated using numerous solvents: aprotic solvents, anhydrous and polar solvents, e.g. dimethyl sulfoxide (DMSO) [18], dimethylformamide (DMF) [18], N-methyl-2-pyrrolidone (NMP) [19,20] and N-cyclohexyl-2-pyrrolidone (CHP) [16], have produced the most stable and uniform dispersions. Overall, LPE yields phosphorene with small thicknesses reaching the monolayer limit.

# 2. Experimental procedures

## 2.1. Preparation of black phosphorus

Black phosphorus (BP) was prepared according to our previous work from red phosphorus by a transport reaction [21]. The chemical vapour transport (CVT) was improved to obtain BP crystals with high purity and crystallinity. A mixture of copper powder from Aldrich (22.75 mg, 99.5%), tin powder from Aldrich (42.5 mg, 99%), red phosphorus lump from Acros Organics (155 mg, 99.99%) and $SnI_4$ from Aldrich (10.0 mg) was placed and sealed under vacuum inside a 10 cm long silica glass ampoule, with a 0.2 cm thick wall and a 1.0 cm inner diameter. The ampoule was positioned inside an oven in a horizontal direction, and the temperature was kept at 923 K for a duration of 4 h. The starting materials were positioned in the high-temperature part of the oven, and the temperature dropped down to ambient temperature for a 3 day period at the rate of 0.2°C min$^{-1}$. The slow-rate cooling provides a better-quality growth. The BP product is synthetized in the cold part of the silica glass ampoule. Minor parts of $SnI_4$ that might remain in the BP crystals are suppressed by putting the product in boiling toluene (ultrasonic bath for 20–45 min) until the toluene remains clear. Figure 1a shows a photograph of synthesized black phosphorus. In order to illustrate the morphological properties of the synthesized black phosphorus, an SEM analysis (figure 1b) allows to observe that the as-grown black phosphorus has a stacked-layer structure thanks to the van der Waals forces, and, therefore, it will be easy to peel each layer off.

## 2.2. Exfoliation

LPE method [19,22–24] has been used to obtain few layers (phosphorene). This technique includes the sonication or shearing [25,26] of stacked-layer crystals in solvents, and it has successfully been applied to graphene materials, transition metal oxide (TMOs) and TMDCs [27–29]. Several solvents were proven to be useful for exfoliation, like amide solvents; for instance: NMP, CHP [30] and isopropanol (IPA). In this work, NMP is used. Figure 2a shows the exfoliation protocol of bulk BP in NMP with a 5 mg ml$^{-1}$ concentration using an ultrasonic bath for 12 h. The temperature is kept at 303.15 K. We explored a serial centrifugation speed to obtain few layers with uniform size and thickness. Figure 2b

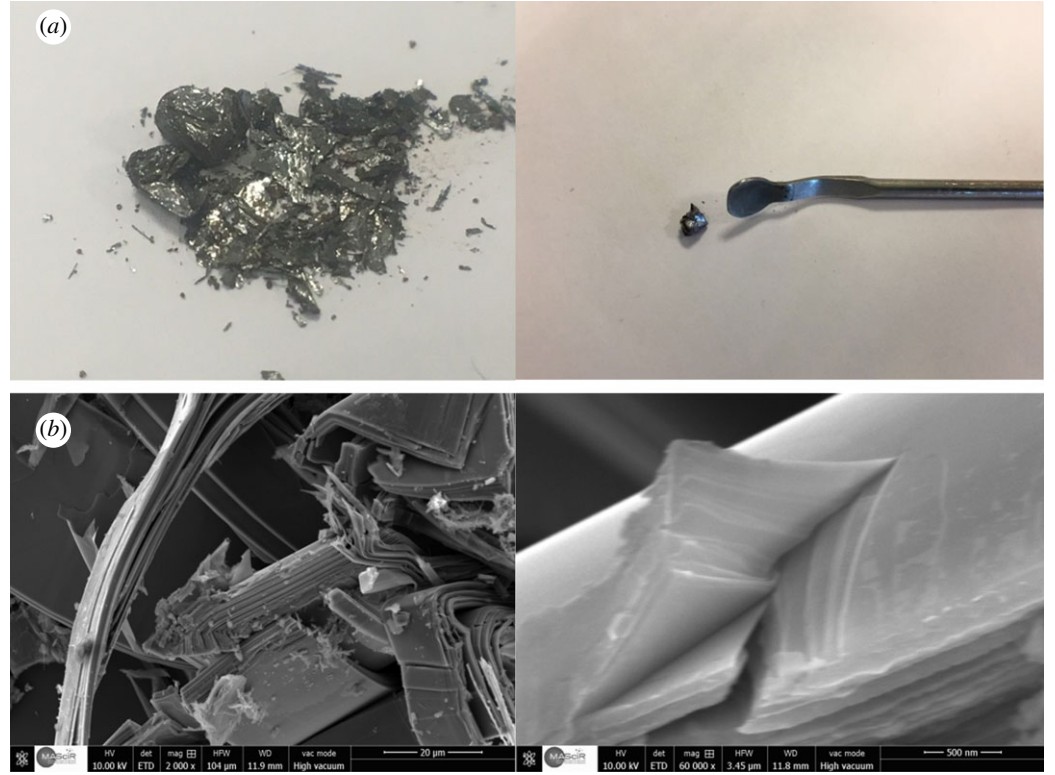

**Figure 1.** (a) Photographs of black phosphorus obtained using the chemical vapour transport growth method. (b) SEM images of synthesized black phosphorus.

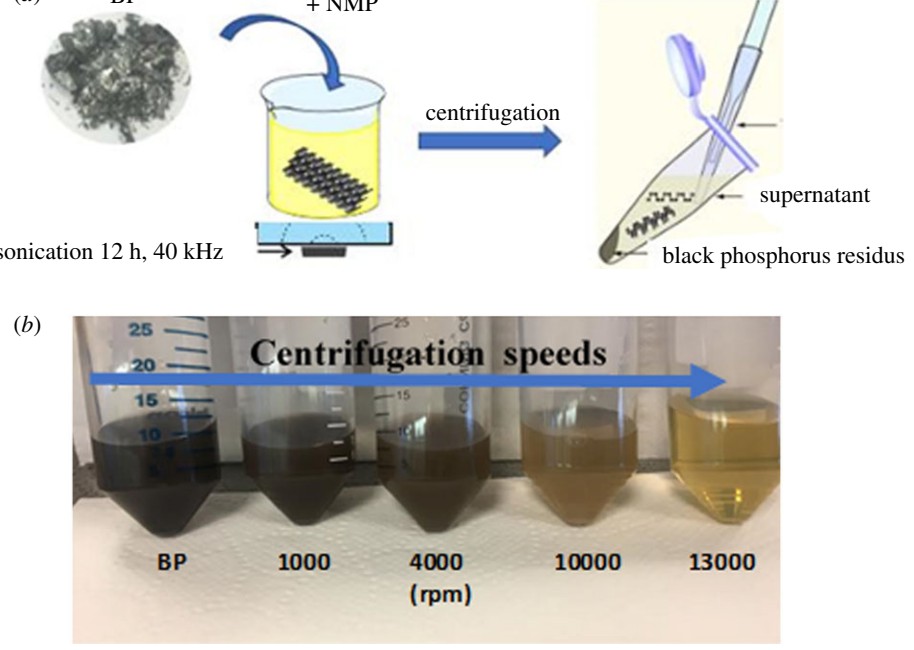

**Figure 2.** (a) An illustration showing the exfoliation process. (b) Dispersions BP and exfoliated BP in NMP centrifuged at different speeds.

shows the dispersions of BP after centrifugation from 1000 to 13 000 r.p.m. range. Some studies of environmental stability of the fabricated nanosheets found that air moisture is absorbed on their surface due to a high hydrophilic character of the few-layer BP, and it has been reported that long-term exposure to ambient conditions degrades the BP, but the few-layer nanosheets remain stable for

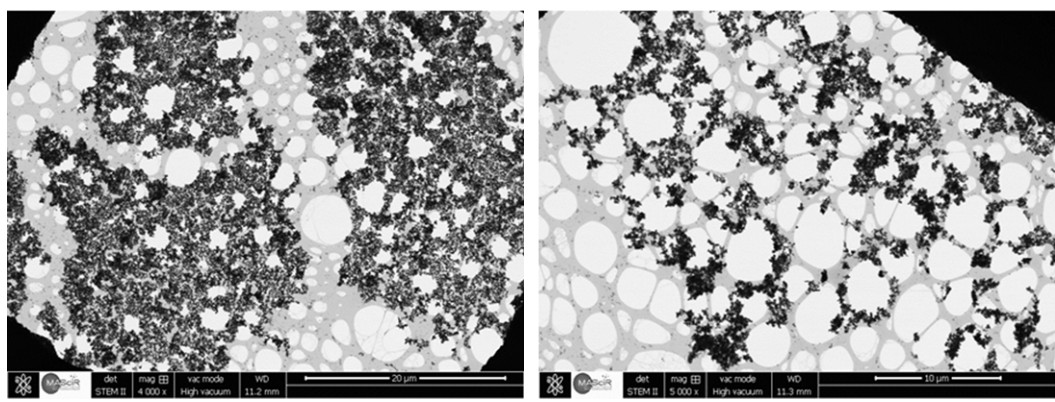

**Figure 3.** TEM images of BP sheets. Inset: lateral size of exfoliated black phosphorus.

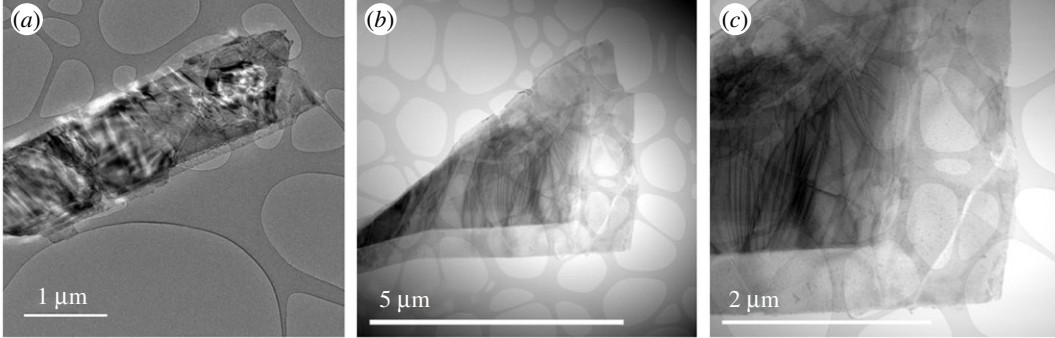

**Figure 4.** TEM images of non-exfoliated BP sheets (*a*), and exfoliated BP sheets after 1 h sonication and 8000 r.p.m. centrifugation (*b,c*).

several days [31]. In our exfoliation process, anhydrous solvents were only opened in an Ar glove box to minimize $O_2$ and $H_2O$ contamination.

# 3. Results and discussion

## 3.1. Characterization of few layers phosphorene exfoliated from as-synthesized black phosphorus

### 3.1.1. Scanning transmission electron microscopy

To confirm the successful exfoliation of synthesized BP in NMP, scanning transmission electron microscopy (STEM) was conducted. The suspended few layers BP were dropped on a copper TEM grid covered with lacey carbon from Ted Pella. The samples were dried at ambient temperature for 48 h under vacuum due to complications in completely eliminating the NMP. The STEM images presented in figure 3 show the morphology and distribution of nanosheets in a dispersion centrifuged at 13 000 r.p.m. Figure 4 shows TEM images of non-exfoliated BP sheets, and exfoliated BP sheets after 1 h sonication and 8000 r.p.m. centrifugation. The size and the morphology of the shown flakes are typical. The lateral size of fabricated phosphorene was approximately 200.8 nm, as shown in the inset of figure 3. It is also important to highlight that the obtained sheets are significantly larger than what is usually detected for other 2D materials [27,30,32,33].

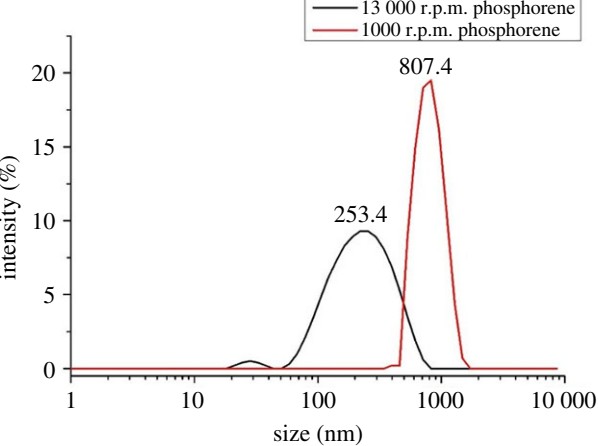

**Figure 5.** Size distribution of centrifuged suspension of black phosphorus at 1000 and 13 000 r.p.m. determined by dynamic light scattering.

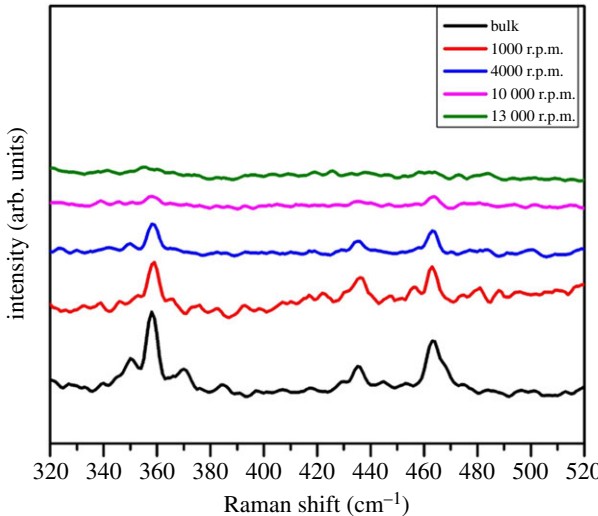

**Figure 6.** Raman spectra of bulk phosphorus and the nanosheets dispersions also measured and plotted for comparison after the subtraction of the *N*-methylpyrolidone signal.

### 3.1.2. Dynamic light scattering characterization and Raman spectroscopy

The light scattering shows that 98% of the dispersion obtained after centrifugation at 1000 r.p.m. is composed of few layers of phosphorene with a lateral size of approximately 800 nm. The supernatants were further centrifuged at 13 000 r.p.m. to obtain 96% of atomically thin BP nanosheets with a lateral size of approximately 253 nm. The lateral dimensions of the phosphorene sheets agree well with the STEM images (figure 5), and are carefully collected and retained for use. This representative dispersion was explored to demonstrate the possibility of exfoliating bulk BP into atomically thin dispersions in NMP. The solvent was later evaporated under vacuum at ambient temperature.

Raman spectroscopy was used to characterize the dispersion of black phosphorus centrifuged at different speeds. The Raman spectrum of the NMP was then subtracted from the spectra recorded for the various analysed solutions and the resulting spectra are shown in figure 6. Three prominent peaks can be ascribed to the phonon modes $A_{1g}$ at 358 cm$^{-1}$, $B_{2g}$ and $A_{2g}$, at 436.7 and 463.1 cm$^{-1}$, respectively. The three Raman bands of the black phosphorus decrease in intensity as the 'r.p.m.' value increases. The signal is almost undetected for the '10 000 r.p.m.' sample and is not detected for the '13 000 r.p.m.' sample. This decline is owing to the thin thickness and small lateral dimensions.

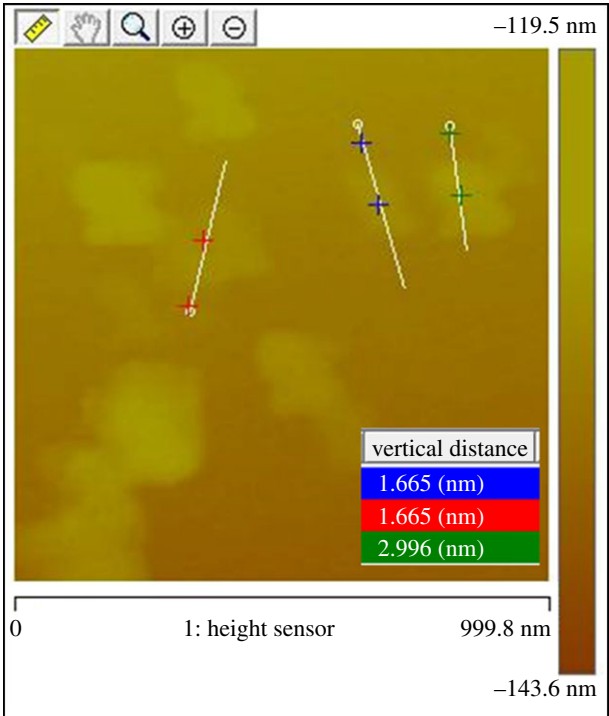

**Figure 7.** AFM image of exfoliated BP layers deposited on SiO$_2$/Si by spin coating process.

### 3.1.3. Atomic force microscopy

In order to investigate the morphology of the surface and the thickness of layers, we have analysed the exfoliated black phosphorus using atomic force microscopy (AFM). A size distribution is obtained after sonication and centrifugation at different speeds. For this work, the supernatant of the dispersion centrifuged at 15 000 r.p.m. is deposited on an Si/SiO$_2$ (001) substrate and spin-coated at 6000 r.p.m. for 1 min. The NMP is evaporated from the substrate under vacuum in ambient temperature at least for 48 h. A typical AFM image of thin-film BP showed the existence of several shapes and sizes of phosphorene with a surface area of approximately 1.6 µm$^2$ (figure 7). The as-synthesized BP sheets had an average thickness of 1.3–2.9 ± 0.9 nm, as shown in figure 7. Previous AFM measurements in the literature [16,34–38] have found that the single-layer phosphorene is 0.9 nm thick. The AFM results indicate that the larger nanosheets consist mostly of two to three phosphorene layers. However, the single or bilayer phosphorene with a thickness from 0.9 to 1.6 nm can be obtained by improving the process conditions. AFM results suggest that increasing both the exfoliation duration beyond 12 h and the speed of centrifugation breaks down the bigger sheets to smaller ones with mainly monolayer phosphorene. Still, we have shown by liquid exfoliation that our synthetic product is easily exfoliable leading to a high quality of layers.

### 3.2. Optical absorption: measurement and Tauc analysis

We have also analysed the ultraviolet (UV)–visible absorption of bulk and dispersions of nanosheets centrifuged at 13 000 and 15 000 r.p.m. to investigate the optical properties of the produced nanosheets, as seen in figure 8a. The obtained suspensions of black phosphorus were also considerably different in their appearances: in transmitted light, a brown colour was detected in diluted suspensions of thick pieces, while those containing nanosheets appeared as yellow transparent liquid in quartz cuvettes.

For the bulk and the suspension centrifuged at 13 000 and 15 000 r.p.m., the optical absorbance spectra were used to draw the Tauc plot as shown in figure 8b. $(\alpha h v)^n$ and the photon energy ($h v$) were linearly dependent, where $\alpha$ stands for the coefficient of absorption, $n$ describes the nature of transition and $h v$ stands for the photon energy. Such linear relationship confirms a direct band gap that is typical of black phosphorus. The Tauc investigation of the transition at high energy shows that the bulk material yields a 1.95 eV transition energy, which increases to 3.2 eV in the bilayer material [12]. These variations obey a power-law curve, and seem to be determined by the quantum

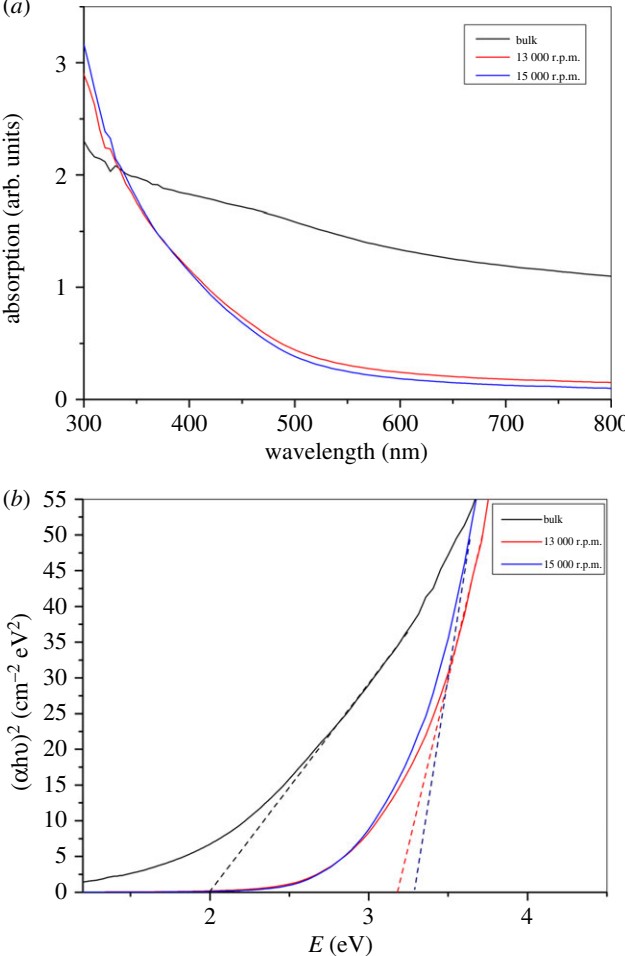

**Figure 8.** (*a*) UV/visible absorption spectrum of dispersions of BP in NMP. (*b*) Representative direct Tauc plots of bulk and centrifuged black phosphorus at 13 000 and 15 000 r.p.m.

**Table 1.** Band-to-band transitions in BP. Adapted from [12].

| layer numbers | band gap energy (eV) | transition at high energy (eV) |
|---|---|---|
| 2 | $1.88 \pm 0.24$ | $3.23 \pm 0.39$ |
| 3 | $1.43 \pm 0.28$ | $2.68 \pm 0.32$ |
| 4 | $1.19 \pm 0.28$ | $2.44 \pm 0.27$ |
| 10 | $0.73 \pm 0.23$ | $2.09 \pm 0.13$ |
| 15 | $0.62 \pm 0.20$ | $2.03 \pm 0.09$ |
| 20 | $0.56 \pm 0.18$ | $2.01 \pm 0.07$ |
| bulk | $0.33 \pm 0.02$ | $1.95 \pm 0.06$ |

confinement [39]. The transition at high energy happens at the Z-point or somewhere near it in the Brillouin zone, and corresponds to a change between either the next highest occupied band (VB − 1) and the conduction band (CB), or the valence band (VB) and the next lowest empty band (CB + 1) [12]. The transition energy in the bulk material is related to earlier band structure measurements of bulk BP [40,41]. In earlier studies, the calculations have shown a relationship between this high energy and the corresponding band gap according to the number of layers, as summarized in table 1 [12]. These data show that the band gap is tuned from 0.33 eV in bulk to 1.88 eV in bilayer material, giving evidence that the band gap and the high-energy transitions are subjected to extreme changes as nanosheets approach the monolayer thickness.

From these results, BP has the potential to offer a new alternative for the design of photocatalysts, solar cells, photodetectors, batteries, transistors as well as the development of novel applications in new fields such the terahertz technology [42–50].

# 4. Conclusion and outlooks

In this work, the LPE was used to produce two to three layers phosphorene from as-synthesized BP. Besides our previous work, this work provides a complete process that starts with the synthesis of BP from red phosphorus, and produces exfoliated few layers of phosphorene. LPE overlays a promising way to mass-produce phosphorene. Many recent reports highlight the use of several solvents and other organic reagents. Based on this work's method, the availability of many organic reagents such as NMP or DMF, combined with the tuning of process conditions such as the centrifugation speed, will provide highly crystalline BP from mono- to multi-layers. Another merit of the liquid exfoliation is the possibility of further isolating the as-dispersed nanosheets from air using the exfoliation solvents, thus slowing down the degradation process significantly. Besides, the ultrafast nonlinear and linear optical properties confirmed by UV–vis–NIR and the Tauc plot analysis showed that BP can cover the band-gap range from 0.33 eV (bulk) to 1.88 eV (bilayer) and the spectrum span between visible and infrared radiation. From these results, there is a great potential for BP as a good alternative nanomaterial for the design of photocatalysts, solar cells, photodetectors, batteries, transistors as well as the development of novel applications in new fields such as the terahertz technology.

Data accessibility. The datasets supporting the results presented in this article are uploaded and available online from the Dryad Digital Repository: https://dx.doi.org/10.5061/dryad.3ffbg79g4 [51].

Authors' contributions. All authors made substantial contributions to this paper. A.E.M., A.E.K and O.M. conceived/planned the experiments, and supervised the findings of this work. G.T., M.A.A, A.B., M.H. and A.L. contributed to the material synthesis and carried the experiments. A.K, A.E., A.M, F.B. and H.O. verified the experimental results and contributed to the interpretation of results, A.E.M and O.M wrote the manuscript with insights from all the authors. A.E.M. secured the funds for this research. All authors discussed the results and contributed to the final manuscript. The corresponding authors of this paper are: A.E.M. and O.M.

Competing interests. The authors declare no conflict of interest.

Funding. This work was partly supported by funds from UAE University Startup project no. 31N312 and UPAR project no. 31N393. A.M. acknowledges the Walloon region for their Beware Fellowship Academia 2015-1, RESIBAT no. 1510399.

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
