## [Reviewer comments · Royal Society Open Science]

Review History

RSOS-201210.R0 (Original submission)

Review form: Reviewer 1

Is the manuscript scientifically sound in its present form?

Yes

Are the interpretations and conclusions justified by the results?

Yes

Is the language acceptable?

Yes

Do you have any ethical concerns with this paper?

No

Have you any concerns about statistical analyses in this paper?

No

Recommendation?

Accept with minor revision (please list in comments)

Comments to the Author(s)

In this work, the authors reported the preparation of phosphorene via liquid exfoliation method using NMP as the solvent and examined the obtained 2D material with analytical tools such as SEM, TEM, Raman, AFM and light absorption. Their results seem highlighting the effective synthesis of high-quality phosphorene.

1- Why NMP was chosen as a solvent in this work, as opposed to other known solvents?

2- Can you comment on how reducing the number of layers increases the bandgap energy?

3- In figure 7 (a), the UV/vis of the 13000rpm is closer to the one of the bulk BP compared to 15000rpm. can you comment?

Review form: Reviewer 2

Is the manuscript scientifically sound in its present form?

Yes

Are the interpretations and conclusions justified by the results?

No

Is the language acceptable?

Yes

Do you have any ethical concerns with this paper?

No

Have you any concerns about statistical analyses in this paper?

No

Recommendation?

Accept with minor revision (please list in comments)

Comments to the Author(s)

See Appendix A.

Decision letter (RSOS-201210.R0)

Dear Dr El Moutaouakil:

Title: Efficient production of few-layer black phosphorus by liquid-phase exfoliation
Manuscript ID: RSOS-201210

Thank you for submitting the above manuscript to Royal Society Open Science. On behalf of the Editors and the Royal Society of Chemistry, I am pleased to inform you that your manuscript will be accepted for publication in Royal Society Open Science subject to minor revision in accordance with the referee suggestions. Please find the reviewers' comments at the end of this email.

The reviewers and handling editors have recommended publication, but also suggest some minor revisions to your manuscript. Therefore, I invite you to respond to the comments and revise your manuscript.

Because the schedule for publication is very tight, it is a condition of publication that you submit the revised version of your manuscript before 11-Sep-2020. Please note that the revision deadline will expire at 00.00am on this date. If you do not think you will be able to meet this date please let me know immediately.

Kind regards,
Dr Ellis Wilde
Publishing Editor, Journals

On behalf of the Subject Editor Professor Anthony Stace and the Associate Editor Dr Dattatray Late.

RSC Subject Editor
Comments to the Author:
Accept with minor revisions

RSC Associate Editor
Comments to the Author:
(There are no comments.)

Reviewer comments to Author:
Reviewer: 1

Comments to the Author(s)

In this work, the authors reported the preparation of phosphorene via liquid exfoliation method using NMP as the solvent and examined the obtained 2D material with analytical tools such as SEM, TEM, Raman, AFM and light absorption. Their results seem highlighting the effective synthesis of high-quality phosphorene.

- 1- Why NMP was chosen as a solvent in this work, as opposed to other known solvents?
- 2- Can you comment on how reducing the number of layers increases the bandgap energy?
- 3- In figure 7 (a), the UV/vis of the 13000rpm is closer to the one of the bulk BP compared to 15000rpm. can you comment?

Reviewer: 2
Comments to the Author(s)
Please see attached

Author's Response to Decision Letter for (RSOS-201210.R0)

See Appendices B & C.

RSOS-201210.R1 (Revision)

Review form: Reviewer 1

Is the manuscript scientifically sound in its present form?

Yes

Are the interpretations and conclusions justified by the results?

Yes

Is the language acceptable?

Yes

Do you have any ethical concerns with this paper?

No

Have you any concerns about statistical analyses in this paper?

No

Recommendation?

Accept as is

Comments to the Author(s)

Authors have addressed my comments and provided with concise and comprehensive answers.

Decision letter (RSOS-201210.R1)

Dear Dr El Moutaouakil:

Title: Efficient production of few-layer black phosphorus by liquid-phase exfoliation

Manuscript ID: RSOS-201210.R1

It is a pleasure to accept your manuscript in its current form for publication in Royal Society Open Science. The chemistry content of Royal Society Open Science is published in collaboration with the Royal Society of Chemistry.

On behalf of the Subject Editor Professor Anthony Stace and the Associate Editor Dr Dattatray Late.

RSC Associate Editor:
Comments to the Author:
Accept as is

RSC Subject Editor:
Comments to the Author:
(There are no comments.)

Reviewer(s)' Comments to Author:
Reviewer: 1

Comments to the Author(s)
Authors have addressed my comments and provided with concise and comprehensive answers.

Appendix A

Manuscript Title : Efficient production of few-layer black phosphorus by liquid-phase exfoliation

Manuscript ID :RSOS-201210

“Efficient production of few-layer black phosphorus by liquid-phase exfoliation” by Ghassane Tiouitchi et al. submitted to Royal Society Open Science.

This paper focus on the efficient production of few layer black phosphorous (BP) by liquid phase exfoliation technique. Authors have tried to produce BP nanosheets in large quantity with uniform size and same thickness using NMP solvent by this technique. I think, this paper is potentially suited for publication in RSOS after some changes in the manuscript.

Here, I would like to share some points which should be addressed in the manuscript.

1. Authors need to mention about stability of few layers BP as it is air sensitive. Also, they should add some important papers on large production for comparison.
2. Ghassane Tiouitchi et al. should provide high-resolution TEM images, which will give information of few layer BP more accurately.
3. Typo errors should be corrected. (Row 41,130,131 and 148).
4. References are not in chronological order in the main text.(for.e.g. references 36, 37 are given before 31-35).
5. Graph formats are not uniform, and some images are blurred. Authors need to modify.
6. Height profile of exfoliated BP in AFM image need to be provided.
7. Authors need to do major changes in the section 3.2. (Optical absorption: measurement and Tauc analysis) as description given in this section is not justifying with the plots 7 (a and b). and with the table 1. They need to put optical spectra of few layer BP of all indicated rpm.
8. The details appearing in the references not as per the format led down by the RSOS.
9. It is not necessary to add references in conclusion.

Appendix B

10/09/2020
Prof. Anthony Stace
Subject Editor
Royal Society Open Science

Dear Subject Editor,

First, we would like to thank you and the referees for your time and your constructive feedback, and for your decision to accept our manuscript titled “Efficient production of few layers black phosphorus by liquid phase exfoliation,” pending the minor revisions.

The manuscript has been reviewed according to the referees’ comments and questions, and all the comments were addressed in the attached file named: "Response to Referees."

We look forward to hearing from you at your earliest convenience.

Sincerely,

Dr. Amine El Moutaouakil
Co-corresponding author
College of Engineering
UAE University
P.O. Box No. 15551, Al Ain, UAE
T: 03-7136575 F:
Email: a.elmoutaouakil@uaeu.ac.ae

Prof. Omar Mounkachi
Co-corresponding author
Laboratory of Condensed Matter and Interdisciplinary Sciences (LaMCSsI),
B.P. 1014, Faculty of Science-Mohammed V University, Rabat, Morocco.
Email: omar.mounkachi@um5.ac.ma

Appendix C

Response to Referees

Manuscript Title : Efficient production of few-layer black phosphorus by liquid-phase exfoliation
Manuscript ID :RSOS-201210

Review 1

- 1- Why NMP was chosen as a solvent in this work, as opposed to other known solvents?
Thank you for the question. Previous graphene studies have found that the choice of organic solvent is critical for efficient solvent exfoliation, with NMP and dimethylformamide (DMF) working particularly well due to their relatively high boiling points and surface tension (~40 mJ/m²). Among conventional solvents, n-methyl-pyrrolidone (NMP) is found to provide stable, highly concentrated dispersions. Additionally, residual NMP from the liquid-phase processing suppresses the rate of BP oxidation in ambient as highlighted in reference 19 of the manuscript: “Kang J, Wood JD, Wells SA, Lee J-H, Liu X, Chen K-S, Hersam MC. 2015 Solvent Exfoliation of Electronic-Grade, Two-Dimensional Black Phosphorus. ACS Nano 9, 3596–3604. (doi:10.1021/acsnano.5b01143)”
- 2- Can you comment on how reducing the number of layers increases the bandgap energy?
Thank you for the question. In the case of very thin film in the order of few interatomic distances, there will be a quantum mechanical confinement of the electrons inside the material leading to the quantization of electronic energy inside the conduction band and valence band. The consequence of this quantization is the increase of the bandgap energy compared to that for the bulk materials. Therefore, there is a direct effect of the size of the material on its band structure; such as when the number of layers decreases, its energy bandgap increases and may turn from indirect to direct bandgap.
- 3- In figure 7 (a), the UV/vis of the 13000rpm is closer to the one of the bulk BP compared to 15000rpm. can you comment?
Thank you for the question. The reason for bringing the two UV/vis spectra together is due to the fact that the difference is not enormous in the number of nanosheets exfoliated at 13,000 and 15,000 rpm.

Review 2

1. Authors need to mention about stability of few layers BP as it is air sensitive. Also, they should add some important papers on large production for comparison.
Thank you for the comment and suggestion. The paragraph below was added in line 86 of the manuscript:

“Some studies of environmental stability of the fabricated nanosheets found that air moisture is absorbed on their surface due to a high hydrophilic character of the few-layer BP, and it has been reported that long-term exposure to ambient conditions degrades the BP, but the few-layer nanosheets remain stable for several days [31]. In our exfoliation process, anhydrous solvents were only opened in an Ar glove box to minimize O₂ and H₂O contamination.”

The following reference has been also added as [31]: “A. Castellanos, «Isolation and characterization of few-layer black phosphorus,» 2D Materials, vol. 1, n° 120531583, 2014.”

2. Ghassane Tiouitchi et al. should provide high-resolution TEM images, which will give information of few layer BP more accurately.

Thank you for the comment and suggestion. We included in the manuscript the TEM images below as Figure 4, to show the non-exfoliated BP, and the exfoliated BP after 1-hour sonication and 8000-rpm centrifugation. Higher centrifugation leads to less clarity in our TEM setup.

Figure 4. TEM images of non-exfoliated BP sheets (left), and exfoliated BP sheets after 1-hour sonication and 8000-rpm centrifugation (middle and right).

3. Typo errors should be corrected. (Row 41,130,131 and 148).
Thank you for your comment and we apologize for the typos. All typos were corrected in the manuscript.
4. References are not in chronological order in the main text.(for.e.g. references 36, 37 are given before 31-35).
Thank you for your comments, and we apologize for the confusion. All references are now ordered as per the order they are cited in the manuscript.
5. Graph formats are not uniform, and some images are blurred. Authors need to modify.
Thank you for your comments, and we apologize for the non uniformity and blurry images. We edited the graphs accordingly.
6. Height profile of exfoliated BP in AFM image need to be provided.
Thank you for your comments, and we apologize for the missing data. We added the height profile in the AFM image.
7. Authors need to do major changes in the section 3.2. (Optical absorption: measurement and Tauc analysis) as description given in this section is not justifying with the plots 7 (a and b). and with the table 1. They need to put optical spectra of few layer BP of all indicated rpm.
Thank you for your comment and suggestion. The section 3.2 was edited to add the suspension centrifuged at 13000 rpm and to justify the plots accordingly.
8. The details appearing in the references not as per the format led down by the RSOS.
Thank you for your comment and we apologize for the style error. All references are now listed as per RSOS format.
9. It is not necessary to add references in conclusion.
Thank you for your comment. All references were taken off the conclusion.